# Zero Waste Management Behavior: Conceptualization, Scale Development and Validation—A Case Study in Turkey

**Sezen Coskun** 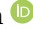

Egirdir Vocational School, Isparta University of Applied Sciences, Egirdir TR32500, Isparta, Turkey; sezencoskun@isparta.edu.tr

**Abstract:** Due to increasing demand on earth sources in all areas, some materials have come under pressure for effective recovery and reuse. In this sense, the management of waste materials has become an important need for effective utilizations. In this regard, the waste management behaviour of individuals towards zero waste was studied using a scale and included pre-testing and administering a survey, and reducing the number of items with the determination of factors. The scale was evaluated using all necessary statistical measures. The IBM SPSS and IBM SPPS AMOS were utilized for confirmatory and expository factor analyses, respectively. It was found that the Cronbach's alpha coefficient determined the reliability level of the improved scale, at 0.909, while the Kaiser–Meyer–Olkin coefficient was determined as 0.887. The Bartlett's sphericity test result was found to be $p < 0.000$. The test results clearly indicated that the sample size was adequate for the measurement of the construct and a patterned relationship among the items was detected. However, the reliability and validity of the developed scaled were confirmed by the goodness of fit indices used. It is important to note that education, profession, level of income, and place of residence significantly influenced the participants' zero waste management behaviour, but the gender and age of the participants were not influential factors. By having these experimental results, it is suitable to suggest that a model consisting of three factors (knowledge, facilities, and motivation) was capable of measuring the waste management behaviour of people towards zero waste in Turkey.

**Keywords:** scale development; survey; zero waste



## 1. Introduction

The increase in the world's population, technological developments, economic growth, and rapid urbanization, have significantly shifted lifestyles and thus consumption habits, which has led to a major increase in solid waste disposal. It has already been well documented that environmentally sustainable consumer behavior is clearly an important aspect of pro-social consumption activities [1]. Instead of spending time and money on waste disposal, the concept of "zero waste" that has emerged in modern waste management is on the way to being implemented worldwide [2,3]. Zero waste management, which aims at the management of resources and waste, requires well-targeted interventions that can help minimize waste. Along with advanced practices for waste recovery/reuse, driving forces (motives) for waste materials and opportunities for improvement should be identified [4]. Otherwise, the issue of zero waste, the implementation of which is aimed as a modern waste management system in urban centers, will remain an unfulfilled quest. There is an enormous demand these days for recycled waste that never existed before [5]. However, one of the most groundbreaking solutions in zero waste management is a pioneering European climate strategy proposed by the European Commission and presented at the 2019 UN COP25 Climate Summit in Madrid, referred to as the European Green Deal [6,7]. In a more recent study conducted by Sahin et al. (2022), it is clearly demonstrated that the driving force is the environmental concern to deal more effectively with solid wastes and to get some of the material out of economic value, where it might be possible to reuse it in some

applications, such as urban space design elements. Numerous researchers have already attempted to create many alternative elements from various type of solid wastes, but the management of waste materials is incomplete and is likely influenced by availability and properties [5].

Pietzsch et al. (2017) claims that scholars have not reached a consensus regarding the concept of ZW [8]. That is why the European Union Commission published the "Towards a Circular Economy: Zero Waste Program for Europe" in July 2014, and created a guide for recycling targets to attain successful zero waste management [9]. It was deemed necessary to publish a document for this purpose and a guide in April 2020, covering zero waste management, medical waste treatment standards, and municipal waste collection services to be applied during the coronavirus crisis, was published [10].

As stated by Bortoleto (2015), waste prevention is an interactive outcome of administrative activities and choices that people make in daily life [11]. Different factors are probably involved in the forming of waste management behavior [12]. Older, richer, and educated people are more likely to be aware of environmental issues and more sensitive to the environment [13]. The presence of recycling facilities is also a contributing factor on waste management behaviors [14]. Improving existing infrastructure facilities over time, and increasing the level of knowledge and awareness of individuals, can enable the faster realization of targets in zero waste management. The interaction of effective waste policies and waste prevention behavior can result in a more sustainable environment.

Studies concerning the development of a scale for the measurement of waste management behavior of people living in Turkey, particularly related to zero waste, are scarce in the literature. In addition, no studies were encountered that quantified which factors might influence zero waste goals. In the international literature, studies aiming to develop a theory of planned behavior [15] to explain the formation of intention and behavior towards resource separation of wastes, and to measure the willingness of residents to pay to realize sustainable waste management, have been identified [16]. In another study conducted in Padang City, Indonesia, with similar modeling, it aimed to understand the social situation of citizens' environmental behavior and to develop a new system suitable for the social situation of Padang city [17]. However, these studies are not related to zero waste modelling. Therefore, the purpose of the present research was to develop a survey scale about zero waste (ZWMS; Zero Waste Management Scale) and to determine the factors affecting the scale. The study design of the research may help to understand the sensitivity of the established zero waste concept. To take advantage of the zero waste topic, the success of zero waste management into valuable secondary material applications could be verified.

## 2. Materials and Methods

### 2.1. Model and Participants in the Study

Participants aged between 15 and 80, living in Turkey, and who were able to respond to the questionnaire through their social media and e-mail accounts, participated in the survey. The research was carried out between 1 January and 1 February 2022, according to the general scanning model. The snowball sampling method was used. A population of 45,000,000 requires a minimum sample size of 384 people (sampling error H = $\pm0.05$ and rates $p$ = 0.5; q = 0.5, $\alpha$ = 0.05) according to the formula given in the literature [18]. The research was completed with 550 participants.

### 2.2. Studies of Scale Development

2.2.1. Item Pool Development

A question pool based on the literature was generated by the investigator to study waste management behaviors towards zero-waste. The questionnaire form consisted of four parts. The demographic characteristics of the participants were established in the first part. In the second part, there were questions to determinate the Factor 1 (F1): "Level of Knowledge" (LK) of participants. The third and fourth sections of the survey were applied to measure other sub-factors, Factor 2 (F2): "Facilities Available" (FA). Factor 3

(F3): "Motivation and Awareness" (MA). The questionnaire was approved by the ethic committee of Isparta University of Applied Sciences on 22 December 2021.

### 2.2.2. Question Pool Validation and Pilot Testing

The preliminary examination of the draft-created items was conducted by experts. The validity of the questions was confirmed by 11 experts chosen from universities. They examined the questions in terms of their scope, content, purpose, and intelligibility. Necessary revisions were made if there was any objection to the items. The survey questions are listed in Table 1. A 5-point Likert scale format was finalized for the 15-item questionnaire to measure the zero waste management behaviors of individuals living in Turkey. A 5-point Likert scale was described by the following: 1 strongly disagree, 2 disagree, 3 neither agree or disagree, 4 agree, 5 strongly agree. Before the main survey, a draft scale was applied to a group of 125 selected participant (randomly) to determine the problems that individuals may have difficulty in understanding.

**Table 1.** Items table.

| Level of Knowledge | |
|---|---|
| Item 1 | I have sufficient knowledge about zero waste management practices. |
| Item 2 | I have enough knowledge about recycling. |
| Item 3 | I have sufficient knowledge about the harmful effects of waste oils on the environment. |
| Item 4 | I have sufficient knowledge about the harmful effects of waste batteries on the environment. |
| Item 5 | I have sufficient knowledge about environmental protection signs (recycling, green dot, ÇEVKO, etc.). |
| **Facilities Available** | |
| Item 6 | I can reach recycling bins where I can throw paper, metal, plastic, glass waste. |
| Item 7 | I can reach the recycling bin where I can throw the waste batteries. |
| Item 8 | I can reach the waste collection point where I can leave vegetable waste oils (frying oil etc.). |
| Item 9 | I have access to the recycling bin where I can leave organic waste (fruit/vegetable peels, leftovers, tea pulp, etc.). |
| **Motivation and Awareness** | |
| Item 10 | I think that zero waste management practice increases public awareness to prevent environmental pollution. |
| Item 11 | I think that the information about the zero waste management application on TV and social media is sufficient. |
| Item 12 | If I reach the recycling bins in my immediate surroundings, I separate my paper, glass, plastic, and glass waste. |
| Item 13 | If I reach the waste oil collection point in my vicinity, I deliver the waste oil I have accumulated. |
| Item 14 | I deliver it to the collection point of electronic waste, such as butteries, old cables, other electronic parts, if I reach it in my close vicinity. |
| Item 15 | The survey study contributed to my zero waste awareness. |

### 2.2.3. Validity Calculation and Reliability

Lawshe's technique was used to determine the content validity of the expert opinions [19]. Frequency, mean, percentage values, and standard deviation, were used to interpret the descriptive statistics. While the Kaiser-Meyer-Olkin (KMO) test was conducted for sample adequacy, Bartlett's test was used to determine suitability for the factor analysis [19]. Exploratory factor analysis (EFA) and confirmatory factor analysis (CFA) were performed using the IBM SPSS Statistics 25.0 and SPSS AMOS 21.0, respectively. EFA and CFA are two common statistical approaches applied in examining the internal reliability of a measure. Internal consistency used in the determination of the reliability level of the developed scale was obtained via the Cronbach's alpha coefficient. EFA was used to define the construct validity of the survey scale. CFA was performed to test the confirmability of the construct resulting from the EFA. CFA was performed to test the construct validity. The composite reliability (CR) and average variance extracted (AVE) values obtained were analyzed for the discriminant validity of the ZWMS.

The conformity of anormal distributed data, such as Likert-type scales to the normal dispersion, can be analyzed with the Q-Q Plot [20]. According to George and Mallery (2010), when the skewness and kurtosis values are between $-2.0$ and $+2.0$, the distribution can be accepted as normal [21]. The zero waste management skewness value is $-0.705$, the kurtosis value is 0.252, the skewness and kurtosis values for a level of knowledge are $-0.623$ and $-0.528$, respectively. The skewness and kurtosis values for the facilities available are 0.269 and $-0.925$, respectively. The skewness and kurtosis value for motivation and awareness are $-1.449$ and 1.754, respectively. Since the scale used showed normal dispersion, statistical evaluations were carried out using parametric tests.

The independent samples *t*-test, F-test (ANOVA), and Bonferroni test, were used to determine whether the ZWMS and its sub-dimensions differ according to the participants' socio-demographic characteristics. The differences between the groups were considered statistically significant if $p < 0.05$.

## 3. Results and Discussion

### 3.1. Demographic Characteristics of the Participants

The results showed that more than 40% of the participants were middle aged (31–45 age of participants), more than 60% were female, mostly lived in cities, and the majority of them had an undergraduate degree. Nearly one quarter of the participants were living on the minimum wage (the minimum wage in Turkey is approximately USD 340) and the rest of them had higher incomes.

### 3.2. Items Validation and Reliability

The calculated content validity index (CVI) and content validity criteria (CVC) were 0.80 and 0.59, respectively, confirming the content validity of the survey. A Cronbach's alpha greater than 0.70 indicates reliability [22]. The confirmation of the self-correlation of the test depends on the Cronbach Alpha Internal Consistency Coefficient [23]. The Cronbach alpha internal consistency coefficients for the reliability level of the developed scale were 0.909. Thus, it can be concluded that scale developed is a reliable and valid instrument to measure the waste management behavior of individuals targeting zero waste (Table 2). The reliability coefficients for the first, second, and third, dimensions were 0.939, 0.848, and 0.898, respectively.

**Table 2.** Item analysis results of the sub-dimensions of ZWMS.

| Items | Total Item Correlation * | t (Lower 27% -Upper 27%) ** |
|:---:|:---:|:---:|
| **ZWMS ($\alpha$ = 0.909)** | | |
| **Level of Knowledge ($\alpha$ = 0.939)** | | |
| LK1 | 0.790 | $-33.294$ *** |
| LK2 | 0.876 | $-36.755$ *** |
| LK3 | 0.859 | $-39.561$ *** |
| LK4 | 0.849 | $-34.042$ *** |
| LK5 | 0.805 | $-35.054$ *** |
| **Facilities available ($\alpha$ = 0.848)** | | |
| FA1 | 0.678 | $-38.600$ *** |
| FA2 | 0.737 | $-44.255$ *** |
| FA3 | 0.715 | $-33.414$ *** |
| FA4 | 0.614 | $-26.385$ *** |
| **Motivation and awareness ($\alpha$ = 0.898)** | | |
| MA1 | 0.619 | $-22.923$ *** |
| MA2 | 0.814 | $-18.210$ *** |
| MA3 | 0.836 | $-21.529$ *** |
| MA4 | 0.759 | $-28.190$ *** |
| MA5 | 0.753 | $-27.143$ *** |
| MA6 | 0.594 | $-19.407$ *** |

* $n$ = 550, ** $n_1$ = $n_2$ = 149, *** significant values for $p < 0.05$.

Table 2 includes independent samples *t*-test results showing the discrimination power of all items. The minimum value required for the item-total test correlation to be sufficient is stated to be 0.20 [24]. The scale items in which we examined the item correlations were above 0.20. The distinctiveness of the items was determined in the ZWMS, the raw scores obtained from each factor were sorted from smallest to largest, and the mean scores of the groups in the lower 27% and upper 27% were compared using the independent sample *t*-test. As a result, there was a significant difference at $p < 0.05$ level for all items for each sub-dimension between the mean scores of the upper and lower group item scores.

### 3.3. Results of the EFA

The KMO value of 0.887 indicates that adequacy was "satisfactory" for factor analysis. Values between 0.5 and 1.0 are considered acceptable as KMO values, while values below 0.5 indicate that factor analysis is not suitable for the dataset in question. A low KMO value means that the indicators are not highly correlated [25] and a threshold value of 0.6 is generally accepted as a threshold value for KMO [26]. Furthermore, the results of the Bartlett's sphericity test were yielding a chi-square value of $\chi2(105) = 6432.065$; $p < 0.01$, which indicated that the items have patterned relationships (Table 3).

**Table 3.** Explanatory factor analysis results for the zero waste management scale.

| Items | Factors | | |
| --- | --- | --- | --- |
| | **Level of Knowledge** | **Facilities Available** | **Motivation and Awareness** |
| LK3 | 0.873 | | |
| LK2 | 0.861 | | |
| LK4 | 0.841 | | |
| LK5 | 0.833 | | |
| LK1 | 0.797 | | |
| FA3 | | 0.838 | |
| FA2 | | 0.794 | |
| FA4 | | 0.789 | |
| FA1 | | 0.763 | |
| MA3 | | | 0.873 |
| MA2 | | | 0.848 |
| MA5 | | | 0.824 |
| MA4 | | | 0.821 |
| MA1 | | | 0.673 |
| MA6 | | | 0.646 |
| **Variance explained (%) (72.925)** | 15.724 | 10.978 | 46.223 |
| Eigenvalues (Λ) | 2.359 | 1.647 | 6.933 |
| **KMO = 0.887; χ2(105) = 6432.065; Bartlett (p) = 0.000** | | | |

The principal component analysis (PCA) with Varimax was used to identify the latent constructs of the ZWMS and reveal the factor pattern. As a result, a model consisting of 15 items was developed with three theoretical dimensions. According to the PCA results, the first factor, LK, consisting of five items, the second factor, FA, including four items, and the third factor, MA, with six items, were obtained. EFA revealed that the loading values were between 0.646 and 0.873. The factor loadings of 0.40 or greater of items were accepted as rational for the construct under examination [27]. The factors used in the scale explain 72.925% of the total variance. The factor loading measures the affiliation of variables to the given factors. According to Tavsancil (2002), the explained variance is between 40% and 60% in multifactorial designs [28].

Based on the results of EFA, the contribution of factors to the total variance was sufficient. As presented in Table 3, the first factor, LK, explained 15.724%; the second factor,

FA, explained 10.978%; and the third factor, MA, explained 46.223% of the total variance. The scree plot in Figure 1 was also helpful for extraction of the factors.

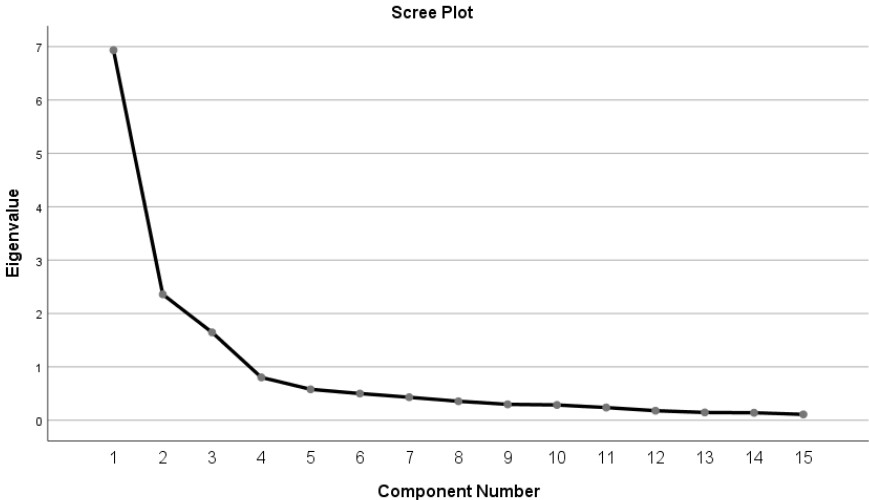

**Figure 1.** Scree plot for the scale developed.

### 3.4. Results of the CFA

Fit indices, which are mostly based on the chi-square value, are used in order to make a decision on the validity of the constructed model. In general, a low chi-square value is evidence of model fit [29]. This indicates that the observed covariance matrix is similar to the predicted covariance matrix. Ullman (2001) suggests that the relative chi-square ($\chi 2/df$) score should be less than 2, while Schumacker and Lomax, (2004), propose a chi-square value of less than 5 [30,31]. The comparative fit indices (CFI) and the root mean square of the approximation (RMSEA) reference values should be 0.95 and 0.08, respectively [32]. Byrne (1994) rejects the goodness of fit index (GFI) scores lower than 0.95 [33]. Incremental fit index (IFI) values exceeding 0.90 are accepted as adequate [34].

The results of the CFA showed that the fit indices of the first model were not acceptable; thus, a modification was applied for correction. While making the modification, the variables that decreased the fit were defined, and new covariances were created for those with high covariance among the values (e1–e2; e8–e9; e13–e14). The modified model had acceptable fit indices (Table 4): RMSEA 0.076; GFI 0.916; IFI 0.958; CFI 0.958; $\chi 2 = 4.195$ (*p* = 0.000).

**Table 4.** Waste management behavior scale goodness of compliance criteria.

| Compliance Indexes | Good Fit | Acceptable Fit | Results before Modification | Results after Modification |
|---|---|---|---|---|
| CMIN/Df | $0 \leq \chi 2/df \leq 3$ | $3 \leq \chi 2/df \leq 5$ | 10.005 | 4.195 |
| GFI | $\geq 0.90$ | $\geq 0.80$ | 0.810 | 0.916 |
| CFI | $0.90 \leq CFI \leq 1.00$ | $0.80 \leq CFI \leq 0.90$ | 0.878 | 0.958 |
| RMSEA | $\leq 0.05$ | $\leq 0.08$ | 0.128 | 0.076 |
| IFI | $\geq 0.95$ | $\geq 0.90$ | 0.878 | 0.958 |

The factor loadings are shown in Table 5, and the model for the first-level confirmatory factor analysis of the waste management behavior scale is shown in Figure 2. As seen in Table 5, factor loads were between 0.56 and 0.93, which comply with the acceptable criteria of the lowest 0.40 [35].

**Table 5.** Factor loads obtained from the confirmatory factor analysis for the zero waste.

| Factors | Factor Loads | CR | AVE |
|---|---|---|---|
| **Level of Knowledge** | | | |
| BD1 | 0.756 | | |
| BD2 | 0.847 | | |
| BD3 | 0.923 | 0.93 | 0.74 |
| BD4 | 0.925 | | |
| BD5 | 0.829 | | |
| **Facilities available** | | | |
| M01 | 0.803 | | |
| M02 | 0.897 | | |
| M03 | 0.690 | 0.83 | 0.56 |
| M04 | 0.563 | | |
| **Motivation and awareness** | | | |
| MF1 | 0.683 | | |
| MF2 | 0.881 | | |
| MF3 | 0.894 | | |
| MF4 | 0.758 | 0.90 | 0.60 |
| MF5 | 0.751 | | |
| MF6 | 0.654 | | |

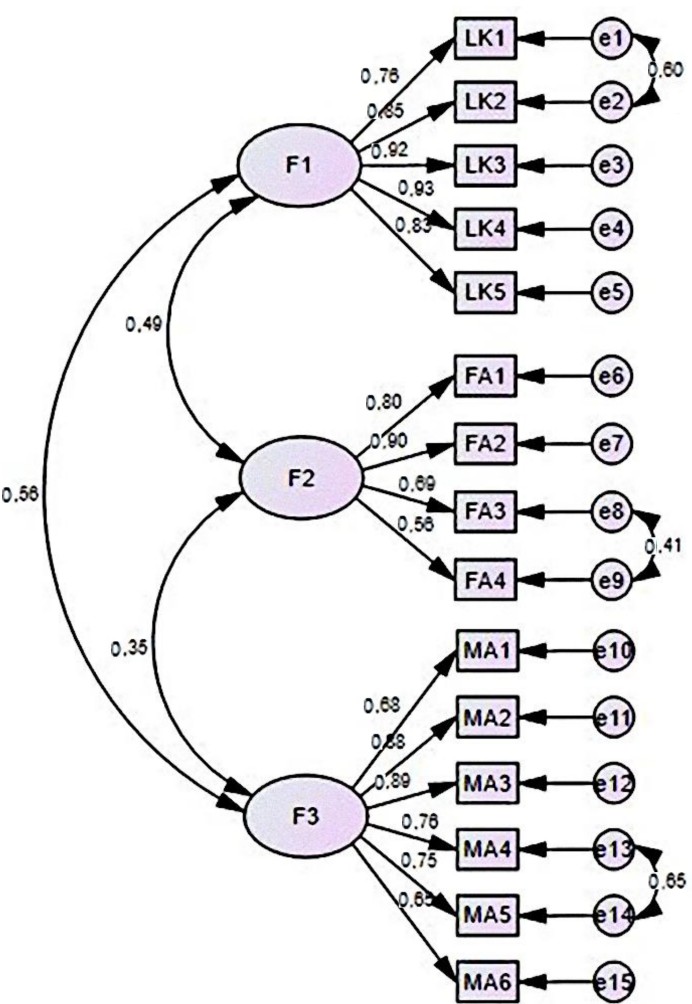

**Figure 2.** Model for first-level multi-factor confirmatory factor analysis of the ZWMS.

The acceptable validity of the scale depends on the CR value of the latent variables in the measurement model being greater than 0.70 and the AVE value being high [36]. As shown in Table 5, both CR and AVE scores of ZWMS (Figure 2) were higher than the thresholds of 0.70 and 0.50, respectively. According to the data obtained, the scale has acceptable validity.

As seen in Figure 2, the factor loadings of MF2 (0.881), MF3 (0.894), MF4 (0.758), and MF5 (0.751), are high. It can be seen from the survey results that the news about zero waste on TV and social media is a suitable method to raise awareness. The fact that recycling bins are available in the immediate vicinity shows that the separate collection of packaging wastes, waste oils, and electronic waste, at the source, will increase the percentages. Considering the MF6 (0.654) factor loading, the survey has greatly contributed to the public's awareness of zero waste and thus it has achieved its purpose.

The study also explored the socio-demographic dimensions of zero waste management behavior by focusing on socioeconomic status, which affects the quality of living and thus the zero waste management behavior of the participants. The independent samples *t*-test, F-test (ANOVA), and Bonferroni test, were used to determine whether the ZWMS and its sub-dimensions differ, based on the socio-demographic characteristics of the participants (Table 6).

**Table 6.** The distribution of scores of the ZWMS and its sub-dimensions by descriptive characteristics of the people.

| Demographic | Level of Knowledge $\overline{X} \pm$ **SS** | Facilities Available $\overline{X} \pm$ **SS** | Motivation and Awareness $\overline{X} \pm$ **SS** | Scale Total $\overline{X} \pm$ **SS** |
|---|---|---|---|---|
| **Age** 18–30 | 17.86 ± 5.36 | 10.74 ± 4.72 | 24.36 ± 5.94 | 52.96 ± 12.47 |
| **Age** 31–45 | 18.16 ± 5.58 | 10.86 ± 4.62 | 24.56 ± 5.19 | 53.59 ± 11.97 |
| **Age** 46–65 | 18.49 ± 5.69 | 10.86 ± 4.62 | 24.85 ± 6.28 | 54.20 ± 13.80 |
| F | 0.486 | 0.044 | 0.278 | 0.372 |
| *p* | 0.615 | 0.957 | 0.757 | 0.690 |
| Female | 17.98 ± 5.56 | 10.85 ± 4.81 | 24.45 ± 5.76 | 53.28 ± 12.82 |
| Men | 18.39 ± 5.48 | 10.77 ± 4.58 | 24.75 ± 5.61 | 53.91 ± 12.18 |
| T | −0.859 | 0.201 | −0.595 | −0.572 |
| *p* | 0.391 | 0.841 | 0.552 | 0.568 |
| **City;** G1 | 18.59 ± 5.40 | 10.67 ± 4.26 | 24.89 ± 4.58 | 54.14 ± 11.16 |
| **City;** G2 | 17.67 ± 5.45 | 11.12 ± 4.85 | 24.56 ± 5.90 | 53.34 ± 12.96 |
| **City;** G3 | 18.36 ± 5.72 | 10.36 ± 4.86 | 24.02 ± 5.85 | 52.74 ± 12.75 |
| **City;** G4 | 18.21 ± 5.65 | 10.85 ± 4.87 | 24.59 ± 6.31 | 53.65 ± 13.28 |
| F | 0.806 | 0.570 | 0.403 | 0.235 |
| *p* | 0.491 | 0.635 | 0.751 | 0.872 |
| Living place: City | 18.37 ± 5.60 | 11.10 ± 4.73 | 24.92 ± 5.42 | 54.40 ± 12.48 |
| Living place: Town | 17.60 ± 5.33 | 10.17 ± 4.64 | 23.75 ± 6.24 | 51.52 ± 12.58 |
| T | 1.514 | 2.143 | 2.226 | 2.484 |
| *p* | 0.131 | 0.033 * | 0.026 * | 0.013 * |
| Edu: Graduate (1) | 19.25 ± 4.97 | 10.95 ± 4.46 | 25.17 ± 4.85 | 55.37 ± 11.27 |
| Edu: Undergraduate (2) | 18.25 ± 5.44 | 10.86 ± 4.78 | 24.77 ± 5.51 | 53.88 ± 12.30 |
| Edu: High school and below (3) | 15.32 ± 6.23 | 10.30 ± 4.78 | 22.04 ± 7.62 | 47.65 ± 15.01 |
| F | 9.949 | 0.405 | 6.587 | 7.707 |
| *p* | 0.000 * | 0.667 | 0.001 * | 0.001 * |
| Bonferroni | 3 < 1; 3 < 2 | - | 3 < 1; 3 < 2 | 3 < 1; 3 < 2 |
| Student (1) | 17.16 ± 5.53 | 10.36 ± 4.46 | 23.91 ± 6.18 | 51.43 ± 12.09 |
| Housewife/retired (2) | 17.43 ± 5.69 | 10.68 ± 4.68 | 23.46 ± 6.44 | 51.58 ± 13.59 |
| Working (3) | 18.24 ± 5.52 | 10.85 ± 4.81 | 24.88 ± 5.38 | 53.96 ± 12.51 |
| Academician/teacher/doctor (4) | 20.30 ± 4.70 | 11.58 ± 4.70 | 25.97 ± 4.64 | 57.84 ± 10.47 |
| F | 4.923 | 0.868 | 3.411 | 4.431 |
| *p* | 0.002 * | 0.458 | 0.017 * | 0.004 * |
| Bonferroni | 1 < 4; 2 < 4; 3 < 4 | - | 2 < 4 | 1 < 4; 2 < 4 |

**Table 6.** *Cont.*

| Demographic | Level of Knowledge $\overline{X} \pm$ **SS** | Facilities Available $\overline{X} \pm$ **SS** | Motivation and Awareness $\overline{X} \pm$ **SS** | Scale Total $\overline{X} \pm$ **SS** |
|---|---|---|---|---|
| **Monthly income** <2500TL (1) | 16.73 ± 5.67 | 9.88 ± 4.74 | 23.34 ± 6.59 | 49.94 ± 13.07 |
| 2501–4000 TL (2) | 18.04 ± 5.85 | 10.92 ± 4.69 | 24.15 ± 6.07 | 53.12 ± 13.71 |
| 4001–6000 TL (3) | 18.88 ± 5.07 | 11.52 ± 4.90 | 25.62 ± 4.98 | 56.02 ± 11.27 |
| 6001–9000 TL (4) | 18.47 ± 5.26 | 10.98 ± 4.47 | 25.11 ± 4.71 | 54.55 ± 11.09 |
| >9000 TL (5) | 19.61 ± 5.09 | 11.11 ± 4.53 | 25.48 ± 4.76 | 56.20 ± 11.37 |
| F | 4.002 | 2.142 | 3.434 | 4.951 |
| *p* | 0.003 * | 0.074 | 0.009 * | 0.001 * |
| Bonferroni | 1 < 3; 1 < 5 | - | 1 < 3 | 1 < 3; 1 < 5 |

* significant values for $p < 0.05$.

The Bonferroni test results also showed that the knowledge level, motivation and awareness, and zero waste management scores, of graduate and undergraduates are higher than those of other participants. Literature findings shown that education, age, gender, and knowledge, are the most influential factors on waste management behavior of people [37,38]; while people's income, education, and employment, have a strong positive impact for sustainable waste management systems, but age and household size factors have a negative impact on the residents' willingness to pay for sustainable waste management [16]. The lack of education diminishes the environmental awareness of rural participants [39]. Zia et al. (2017) stated that waste management was also a process that is more pronounced, and more intense, in spring and winter [40]. A study conducted to explain the formation of intention and behavior towards the resource separation of wastes showed that motivation has the most important effect on intention, followed by moral imperative, perceived behavioral control, subjective norm, situational factor, and attitude [15].

In the Bonferroni test results, monthly income influenced positively and significantly the waste management behavior of participants. Similar results were also reported by Xiao and Zhou (2020), who showed that households with higher income are more sensitive to recycling and protecting the environment from pollution [41]. The results of our study showed that some professions (academician/teacher/doctor) have more knowledge and are more motivated and aware of waste management than other participants ($p < 0.05$). Studies have shown that the residential area of households is a determining factor regarding waste generation [42,43], and there is also significant difference between rural and urban areas, in terms of the amount of waste generated [44]. Similarly, the results showed that city residents are more likely to use waste facilities, are more motivated, and are more aware of zero waste management than those living in rural areas ($p < 0.05$). The test results also indicated that education level is an important factor influencing the waste management behavior of the participants. The participants having a postgraduate education had higher scores than the others ($p < 0.05$).

The study also explored socio-demographic factors and showed that married people score higher than singles, and unmarried women are willing to pay more for energy efficient goods. Other results of the study showed that middle aged people (36–50 age of participants) indicate more environmentalist behavior than young people, and higher education increases sensibility [45].

## 4. Conclusions

Every study design is unique, and each will work best with a different combination of pieces. Thereby, it is important to decide which methodology will work best. It is essential to understand the principles of the zero waste management concept towards effective utilizations. The study does not aim to explain all the waste disposal problems, instead it was aimed to introduce a conceptually valid zero waste management methodology. In this respect, an experimental approach was developed comprised of 15 items to determine the waste management behavior of people towards zero waste in Turkey. The KMO

and Bartlett's tests clearly demonstrated that the 15 items scale had construct validity. However, internal consistency reliability was estimated among the latent construct, the level of knowledge, facilities available, and motivation and awareness towards zero waste. The results proved that the scale is valid and reliable for the study. Although the results indicated that gender and age do not significantly affect zero waste management behavior, education, the level of income, profession, and living place, significantly influenced towards the zero waste concept.

The conducted survey also reveals that news about zero waste on TV and social media is watched carefully by the public, and it may be a suitable method to raise the awareness of the public. The fact that recycling bins are available in the immediate vicinity shows that the separate collection of packaging wastes, waste oils, and batteries, at the source, will increase the percentages, and the public will be motivated in this regard. The survey results have been found to greatly contribute to the public's awareness of zero waste.

**Funding:** This research has not received any funding. APC was funded by the author herself.

**Institutional Review Board Statement:** There is an ethical committee decision regarding the survey (Date: 22 December 2021 Number: E-96714346-050.99-63096).

**Informed Consent Statement:** Informed consent was obtained from all subjects involved in the study.

**Data Availability Statement:** Not applicable.

**Conflicts of Interest:** The author declares no conflict of interest.

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
