# Peer review of "Zero Waste Management Behavior: Conceptualization, Scale Development and Validation—A Case Study in Turkey"

_sustainability, doi:10.3390/su141912654_

Round 1

Author Response

Thank you for your comments. Revisions have been made with precision.

Reviewer 2 Report

-The paper is full of mistakes, specially linguistic and grammatical mistakes

-The novelty of the paper should be highlighted

-The introduction part should be enhanced

-Objective of the research is not clearly written from the gap of research

-This article lacking in terms of critical discussion based on the findings obtained

-The number of references should be limited, of which more than 16% of the paper is references

-The authors should remove any reference does not follow Web of Science (ISI)

Author Response

Thank you for comments. Revisions have been made with precision.

The author had proofreading done by a native speaker and the relevant certificate was attached to the supplementary documents.

In addition, corrections were made with the support of re-editing.   The introduction part should be enhanced.  In addition, the introduction part should be enhanced. References are limited.   Best Regards, 

Reviewer 3 Report

This paper presents survey research about individual behaviors toward zero waste management. The topic of this study is essential for sustainability. However, the current version is more like a survey report to demonstrate the method and result in accordance with the statistical criteria. There are too many specific terms in the current result and discussion, which hinders understanding the results. To improve the value of the contribution to waste management and sustainability, I suggest that a significant revision is mandatory to consider the publication. The statistical test and validation content can be moved to the supporting information to support the rigorousness. More explanation of the survey design (e.g., what a question stands for the impact on zero waste management), the discussion on the results of each question, and the suggestions for improving zero waste management are necessary to be included in depth.

Reviewer 4 Report

The paper is well written and easy to follow. It contins only a necessary data, and data required to understand the basic concept of the work, which contributed by its good structure and good preparation.

I have only few sugdesstions:

The Title of the paper should contain that it is a case study from the Turkey. In the presented for, it is more general and can be considered world wide. Thus "the case study of Tukey" should be added in the manuscript.

Since the Turkey is a big country, we do not have an information where the study was conducted, in which part of the Turkey, or city? It should be added as well. It is not the same if the study was conducted in Istambul, Mardin or a llitlle village near Ankara.

Figure 2 is bllured when printed, thus it should be converted in higher resoultion.

Round 2

Reviewer 1 Report

The authors have made all the proposed changes

Reviewer 2 Report

Most of the previous comments have been addressed

Reviewer 3 Report

No additional suggestions.